# Impact of Experimental Bias on Compositional Analysis of Microbiome Data

**DOI:** 10.3390/genes14091777

**Published:** 2023-09-08

**Authors:** Yingtian Hu, Glen A. Satten, Yi-Juan Hu

**Affiliations:** 1Department of Biostatistics and Bioinformatics, Emory University, Atlanta, GA 30322, USA; yingtian.hu@emory.edu; 2Department of Gynecology and Obstetrics, Emory University School of Medicine, Atlanta, GA 30322, USA; gsatten@emory.edu

**Keywords:** compositional effect, false-discovery rate (FDR), LOCOM, interaction bias, main effect bias, taxon ratios, test differential abundance

## Abstract

Microbiome data are subject to experimental bias that is caused by DNA extraction and PCR amplification, among other sources, but this important feature is often ignored when developing statistical methods for analyzing microbiome data. McLaren, Willis, and Callahan (2019) proposed a model for how such biases affect the observed taxonomic profiles; this model assumes the main effects of bias without taxon–taxon interactions. Our newly developed method for testing the differential abundance of taxa, LOCOM, is the first method to account for experimental bias and is robust to the main effect biases. However, there is also evidence for taxon–taxon interactions. In this report, we formulated a model for interaction biases and used simulations based on this model to evaluate the impact of interaction biases on the performance of LOCOM as well as other available compositional analysis methods. Our simulation results indicate that LOCOM remained robust to a reasonable range of interaction biases. The other methods tend to have an inflated FDR even when there were only main effect biases. LOCOM maintained the highest sensitivity even when the other methods could not control the FDR. We thus conclude that LOCOM outperforms the other methods for compositional analysis of microbiome data considered here.

## 1. Introduction

Experimental bias is a pervasive feature of microbiome data because each step in the sequencing experiment workflow (i.e., DNA extraction, PCR amplification, amplicon sequencing, and bioinformatics processing) favors certain taxa over others [1,2,3]. Absent experimental procedures that produce unbiased data, it is therefore necessary to account for experimental bias when analyzing microbiome data. Fortunately, McLaren, Willis, and Callahan (MWC) [4] have proposed a model explaining how experimental bias affects microbiome data. In this model, the observed relative abundance of each taxon is a product of the taxon’s true relative abundance and a taxon-specific bias factor, normalized over all taxa observed in the sample. Each taxon-specific bias factor represents the accumulation of multiplicative biases over all steps in the experimental pipeline so that multiple sources of bias are described by a single factor. With the MWC model in mind, we developed LOCOM [5], a logistic regression-based compositional analysis method for detecting differentially abundant taxa, that only estimates parameters that are free from bias, i.e., are not affected by bias factors. To the best of our knowledge, LOCOM was the first method that accounted for experimental bias and was shown both analytically and numerically to be fully robust to any bias that follows the MWC model; none of the other existing methods for testing taxon differential abundance have considered experimental bias.

The MWC model assumes no between-taxon “interaction” bias, i.e., the presence or abundance of one taxon does not affect the bias factors of any other taxa in the sample. LOCOM may not be bias-robust if the “main effect” biases of the MWC model were supplemented by interaction biases. In fact, Zhao and Satten [6] investigated the existence of interaction biases by expanding the MWC model to include interaction terms and then fitting this generalized model to mock community samples [7]. They found a small amount of evidence for interaction biases, albeit with smaller magnitudes than the main effect biases in the MWC model. This finding is the motivation for this study. Here, we formulate a new model for generating interaction biases and use simulations based on this model to evaluate the impact of these interaction biases on the performance of LOCOM, as well as a number of existing approaches (ANCOM [8], ANCOM-BC [9], fastANCOM [10], ALDEx2 [11], WRENCH [12], DACOMP (v1.26) [13], LinDA [14], and the Wilcoxon rank-sum test of log-ratio transformed data after adding pseudo counts of either 0.5 or 1). Thus, this report provides the first assessment of the impact of interaction biases on the analysis of microbiome data. Note that this assessment can only be achieved by simulations instead of real data since we cannot control the nature of the bias even using model community data.

## 2. Methods

We generalize the MWC model using the framework of Zhao and Satten [6] by adding an interaction bias $\theta_{jj^{\prime }}$ to the log-linear model:(1)$$\log\left(p_{ij}\right)=\log\left(\pi_{ij}\right)+\gamma_{j}+\sum_{j^{\prime }\neq j}\theta_{jj^{\prime }}\pi_{ij^{\prime }}+\alpha_{i}\;,$$
where $\pi_{ij}$ is the true relative abundance of taxon *j* in sample *i*, $p_{ij}$ is the expected value of the observed relative abundance, $\gamma_{j}$ is the main effect bias for taxon *j* in the MWC model, and $\alpha_{i}$ is the sample-specific normalization factor that ensures the compositional constraint on $p_{ij}$. We followed [6] to introduce the effects of covariates $X_{i}$ on taxon *j* by replacing $\log(\pi_{ij})$ in (Equation 1) with $\log\left(\pi_{j}^{0}\right)+X_{i}^{T}\beta_{j}$, where $\beta_{j}$ contains the effect sizes, and $\pi_{j}^{0}$ is the true relative abundance of taxon *j* when $X_{i}=0$. Note that, similar to all compositional analysis methods, the effect size parameters $\beta_{j}$ here characterize differences in “absolute” abundances rather than “relative” abundances; relative abundances are further modulated by the normalization factor $\alpha_{i}$ which is a function of all the $\beta_{j}$s. The interaction $\theta_{jj^{\prime }}$ determines the extent to which taxon $j^{\prime }$ affects the bias factor for taxon *j*. The assumption underlying the MWC model is $\theta_{jj^{\prime }}=0$ for all *j* and $j^{\prime }$. We adopt the following model for generating the interaction biases relative to the main effect biases in our simulation studies:(2)$$\theta_{jj^{\prime }}=-\mathrm{sign}\left(\gamma_{j^{\prime }}\right)\phi \epsilon_{jj^{\prime }}\left|\gamma_{j}\right|\;,$$
where ϕ is a constant for all taxa pairs that we call the magnitude of interaction biases, and $\epsilon_{jj^{\prime }}$ is a non-negative error term with a mean of one.

The value of ϕ and the minus sign in (Equation 2) reflect the findings of Zhao and Satten [6], who considered a model similar to (Equation 1) but with an interaction that depended on the presence of a taxon rather than its relative abundance. In particular, they considered the model
(3)$$\log\left(p_{ij}\right)=\log\left(\pi_{ij}\right)+\gamma_{j}+\sum_{j^{\prime }\neq j}{\tilde{\theta }}_{jj^{\prime }}I\left[\pi_{ij^{\prime }}>0\right]+\alpha_{i}\;,$$
where ${\tilde{\theta }}_{jj^{\prime }}$ determines the extent to which the *presence* of taxon $j^{\prime }$ affects the bias factor for taxon *j*. Using data from all seven taxa in the Brooks mock community samples, they estimated the main effect biases $\gamma_{j}$s and interaction biases ${\tilde{\theta }}_{jj^{\prime }}$s and presented the results in their Table 4. For the convenience of our readers, we display their results in Figure 1. First, we observe that the interaction biases have smaller magnitudes than the main effect biases; we calculated the mean of the interaction–main effect ratios (i.e., $|{\tilde{\theta }}_{jj^{\prime }}/\gamma_{j}|$) to be 0.204. Because ${\tilde{\theta }}_{jj^{\prime }}$ in model (Equation 3) corresponds to the product $\theta_{jj^{\prime }}\pi_{ij^{\prime }}$ in our model (Equation 1) and because most of the Brooks samples have three taxa with equal proportions, i.e., $\pi_{ij^{\prime }}=1/3$, we obtained the (approximate) mean of $|\theta_{jj^{\prime }}/\gamma_{j}|$ and, thus, an estimate for ϕ of 0.612. In our simulations, we varied the value of ϕ from 0 to 4, which includes our empirical estimate of 0.612 while also extending the range of ϕ to allow for stronger interaction biases than those found in the Brooks data. In addition, Figure 1 shows that the interaction bias in taxon *j* caused by taxon $j^{\prime }$ has an opposite sign to the sign of the main effect bias in taxon $j^{\prime }$, except for some very small interaction biases. This implies that if taxon $j^{\prime }$ has a measured abundance higher than its true abundance, it is associated with decreased measured abundances of other taxa, which seems plausible if we assume that only a relatively constant total number of amplicons are sequenced. Finally, interaction biases are not necessarily symmetric between a pair of taxa, as also noted by Zhao and Satten [6], so our model (Equation 2) also does not guarantee such symmetry.

Our simulation studies were based on models (Equation 1) and (Equation 2) and data on 856 taxa of the upper-respiratory tract (URT) microbiome by Charlson et al. [15]. We considered three scenarios for the distribution of interaction biases. In the first scenario (referred to as S-nondiff), we sampled $\epsilon_{jj^{\prime }}/2$ from $N(0.5,0.1^{2})$ for all taxa pairs *j* and $j^{\prime }$. In the second scenario (referred to as S-diff-causal), we modified S-nondiff to sample $\epsilon_{jj^{\prime }}/2$ from Beta(0.5,0.5) when taxon *j* was “causal” (i.e., associated with the trait of interest) and for all $j^{\prime }$. Both distributions have a mean of 0.5; however, $N(0.5,0.1^{2})$ has one mode at 0.5, whereas Beta(0.5,0.5) has two modes 0 and 1 and, hence, a variance of larger than $0.1^{2}$. In the third scenario (referred to as S-diff-half), we used $N(0.5,0.1^{2})$ for half of the randomly selected taxa *j*s and Beta(0.5,0.5) for the remaining half of the taxa. Unlike S-nondiff, S-diff-causal has a modest (but trait-related) variation in $\epsilon_{jj^{\prime }}$, while S-diff-half has a large variation in $\epsilon_{jj^{\prime }}$ that is not trait-related. The distribution of the bias factor due to taxon–taxon interactions, $\eta_{ij}=\sum_{j^{\prime }\neq j}\theta_{jj^{\prime }}\pi_{ij^{\prime }}$, in each taxon *j* across all samples is displayed in Appendix A and has the following features. In all scenarios and for all taxa *j*s, the mean of $\eta_{ij}$ was approximately zero due to the averaging of contributions with different directionalities. The variance of $\eta_{ij}$ increased as $\gamma_{j}$ increased at a given ϕ. S-nondiff and S-diff-causal had the same $\eta_{ij}$ values at the null taxa and different $\eta_{ij}$ values at the causal taxa. Finally, in S-diff-half, $\eta_{ij}$ had a larger variance in one half of the taxa than in the other half of the taxa.

Additional details of the simulation settings generally follow the simulations in [5]. Specifically, we considered both binary and continuous traits of interest without any confounding covariates; we also considered a binary confounder when the trait was binary. We used the two sets of “causal” taxa (i.e., taxa that are associated with the trait) that were used in [5] and referred to as M1 and M2. In M1, 20 taxa were randomly sampled to be causal from the set of taxa having mean relative abundances in the URT data [15] greater than 0.005 (but excluding the most prevalent taxon). In M2, the top five most abundant taxa (having mean relative abundances of 0.105, 0.062, 0.054, 0.050, and 0.049) were selected to be causal. For simulations with a binary confounder, we assumed that the confounder was associated with 20 taxa under M1 (10 sampled at random from the causal taxa and 10 from the null taxa) and 5 taxa under M2 (2 from the causal taxa and 3 from the null taxa). We set the sample size to 100. Let $T_{i}$ denote the trait and $C_{i}$ denote the confounder for the *i*th sample so that $X_{i}={\left(T_{i},C_{i}\right)}^{T}$. To generate a binary trait, we selected an equal number of samples with $T_{i}=1$ and $T_{i}=0$. When a binary confounder was present, we drew $C_{i}$ from Bernoulli(0.2) in samples with $T_{i}=0$ and from Bernoulli(0.8) in samples with $T_{i}=1$. To generate a continuous trait, we sampled $T_{i}$ from U[−1,1]. To simulate the read count data for the 856 taxa, we first sampled the baseline relative abundances $\pi_{i}^{\left(0\right)}=\left(\pi_{i1}^{\left(0\right)},\pi_{i2}^{\left(0\right)},\ldots ,\pi_{iJ}^{\left(0\right)}\right)$ of all taxa for each sample from $Dirichlet(\bar{\pi },\theta )$, in which $\bar{\pi }$ and θ took the estimated mean and overdispersion (0.02) from fitting the Dirichlet–Multinomial model to the URT data. We formed the *true* relative abundances $\pi_{ij}$ for all taxa by spiking the causal taxon $j^{\prime }$ with an $\exp(\beta_{j^{\prime },1}T_{i})$ fold change, spiking the confounder-associated taxon $j^{\prime \prime }$ with an $\exp(\beta_{j^{\prime \prime },2}C_{i})$ fold change, and normalizing the relative abundances, so that
$$\pi_{ij}=\frac{\exp\left(\beta_{j,1}T_{i}+\beta_{j,2}C_{i}\right)\pi_{ij}^{\left(0\right)}}{\sum_{j^{*}=1}^{J}\exp\left(\beta_{j^{*},1}T_{i}+\beta_{j^{*},2}C_{i}\right)\pi_{ij^{*}}^{\left(0\right)}}\;.$$
We formed the *observed* relative abundances $p_{ij}$ by additionally multiplying the bias factor $\exp(\gamma_{j}+\sum_{j^{\prime }}\theta_{jj^{\prime }}\pi_{ij^{\prime }})$, so that
$$p_{ij}=\frac{\exp\left(\gamma_{j}+\sum_{j^{\prime }}\theta_{jj^{\prime }}\pi_{ij^{\prime }}+\beta_{j,1}T_{i}+\beta_{j,2}C_{i}\right)\pi_{ij}^{\left(0\right)}}{\sum_{j^{*}=1}^{J}\exp\left(\gamma_{j^{*}}+\sum_{j^{\prime }}\theta_{j^{*}j^{\prime }}\pi_{ij^{\prime }}+\beta_{j^{*},1}T_{i}+\beta_{j^{*},2}C_{i}\right)\pi_{ij^{*}}^{\left(0\right)}}\;.$$
Note that $\beta_{j,1}=0$ for null taxa and $\beta_{j,2}=0$ for confounder-independent taxa. For simplicity, we set $\beta_{j,1}=\beta$, a common effect size of the trait, on all causal taxa. We generated the main effect bias $\gamma_{j}$ from $N(0,0.8^{2})$, which gave a range between 0.2 and 5 for most (95%) fold changes ($\exp\gamma_{j}$) caused by the main effect bias. The scheme for generating the interaction bias $\theta_{jj^{\prime }}$ has been described earlier. Finally, we generated the taxon count data using the Multinomial model with the mean $\left(p_{i1},p_{i2},\ldots ,p_{iJ}\right)$ and a library size sampled from an *N*(10,000, (10,000/3${)}^{2})$ distribution that was left-truncated at 2000.

Following [5], we applied the following compositional analysis methods: LOCOM, ANCOM, ANCOM-BC, ALDEx2, DACOMP (v1.26), WRENCH, Wilcox-alr-half, and Wilcox-alr-one. The latter two are Wilcoxon rank-sum tests of additive-log-ratio transformed count data after adding pseudocount 0.5 or 1 to all count data and using the most abundant null taxon (known in simulated data) as the reference; the *p*-values were adjusted for multiple testing by the Benjamini–Hochberg [16] procedure. In addition, we included two newly developed methods, fastANCOM and LinDA. LOCOM requires that the taxa present in fewer than 20% of samples are filtered out. Recall that, since LOCOM fits logistic regression models to taxa count data, zero read counts are naturally handled as possible outcome values. We applied all methods to each replicate of simulated data after using this filter. For ANCOM, ANCOM-BC, fastANCOM, and LinDA, we also applied them to data using their own filter with 10% presence as the cutoff and denoted them as ANCOM*^o^*, ANCOM-BC*^o^*, fastANCOM*^o^*, and LinDA*^o^*. The empirical FDR and sensitivity (proportion of truly causal taxa that were detected) were evaluated at the nominal level of 20% based on 1000 replicates of data. A relatively high nominal FDR level was chosen due to the small numbers of causal taxa in both M1 and M2.

## 3. Results

The empirical FDR and sensitivity of all aforementioned compositional analysis methods for detecting the “causal” taxa (i.e., taxa that are associated with the trait), for simulations with a binary trait and no confounder under scenarios M1 and M2, are displayed in Figure 2 and Figure 3, respectively (results for simulations with confounders and continuous traits showed similar patterns and are therefore deferred to Appendix A). Note that ϕ=0 corresponds to no interaction bias at any taxa (i.e., main effect biases only), and β=0 corresponds to no differential abundance at any taxa (i.e., the global null). In all cases, the FDR of LOCOM remained at or close to the nominal level as long as the magnitude of the interaction bias ϕ<1, which is substantially larger than the empirical value of 0.612 observed in the Brooks data [6]; under the global null in particular, LOCOM always controlled the FDR regardless of the value of ϕ. It was only when both ϕ and β became unrealistically large that we observed moderate inflation in the FDR of LOCOM. The FDR inflation of LOCOM was similar in S-nondiff and S-diff-causal because the interaction biases were similarly distributed at the majority of taxa, which were null taxa; the inflation was (slightly) larger in S-diff-half because the interaction biases had the largest variability among the three scenarios at a given ϕ. The interaction biases caused some loss of sensitivity for LOCOM, but the drop was relatively small, and LOCOM maintained the highest sensitivity among all methods in all of our simulation scenarios. The results of the other methods showed a similar trend to LOCOM with the FDR increasing and the sensitivity decreasing as ϕ increased, as well as an increase in FDR inflation as β increased.

The other methods had very different performances even when there were only main effect biases (ϕ=0). ANCOM-BC and fastANCOM performed the best among the other methods, controlling the FDR for the full range of β values we have considered when ϕ was small and having similar FDR inflation as LOCOM when both β and ϕ became large; however, this performance came at the cost of a substantially lower sensitivity than that of LOCOM. ANCOM had a moderately inflated FDR when β was small but performed better when β was increased; nevertheless, its sensitivity was among the lowest. ALDEx2 had an inflated FDR and poor sensitivity when β was large in M1 but not in M2. WRENCH, Wilcox-alr-half, and Wilcox-alr-one had highly inflated FDRs except in the global null case, and LinDA had a highly inflated FDR in all cases including the global null. These results were based on data after applying the filter of rare taxa as recommended by LOCOM; in general, ANCOM, ANCOM-BC, fastANCOM, and LinDA had worse FDR control when a less stringent filter was applied. Note that, unlike the LOCOM paper containing results from the older version (v1.1) of DACOMP, we used the latest version (v1.26) here, which yielded a highly inflated FDR and low sensitivity in M2 because some causal taxa were incorrectly selected to be among the reference set.

## 4. Discussion

In this study, we found that LOCOM was robust to not only main effect biases but also a reasonable range of interaction biases. The other methods tended to have an inflated FDR even when there were only main effect biases; many of them did not control the FDR even when there was no experimental bias at all (results shown in [5]). LOCOM maintained the highest sensitivity among all methods even when the other methods did not control the FDR. Therefore, we conclude that LOCOM outperforms most (if not all) existing methods for compositional analysis of microbiome data.

The robust performance of all methods to the interaction bias is likely due to each term $\theta_{jj^{\prime }}\pi_{ij^{\prime }}$ in (Equation 1) that governs the effect of taxon $j^{\prime }$ on the bias factor of taxon *j* being small because the relative abundance $\pi_{ij^{\prime }}$ is generally small. Further, even if all contributions from different taxa to the total interaction bias $\sum_{j^{\prime }}\theta_{jj^{\prime }}\pi_{ij^{\prime }}$ have the same sign, any non-zero mean interaction bias across taxa (or individuals) would be automatically included in the normalization factor $\alpha_{i}$ (or the main effect bias $\gamma_{j}$). Thus, the interaction bias is “automatically centered” by these constraints, which may also decrease the apparent effect of the interaction bias.

We used the Brooks mock community data to motivate our simulation studies, which may have two limitations. First, the Brooks samples contain at most seven taxa, all with equal true relative abundance; other mock community datasets, especially datasets that contain a large number of taxa and that mimic real microbiome data, should be used to study the interaction bias. Second, bacteria in a real microbial community may have different interactions from bacteria in a mock community. Therefore, the study of experimental biases and their impact on downstream analysis continues to be an important topic for the foreseeable future.

## 5. Key Points

Microbiome data are subject to experimental bias, which not only takes the form of taxon-specific main effects but also taxon–taxon interactions.LOCOM is robust to all main effect biases and a reasonable range of interaction biases.With the exception of LOCOM, the currently available methods tend to have inflated FDRs even when there are only main effect biases.

## Figures and Tables

**Figure 1 genes-14-01777-f001:**
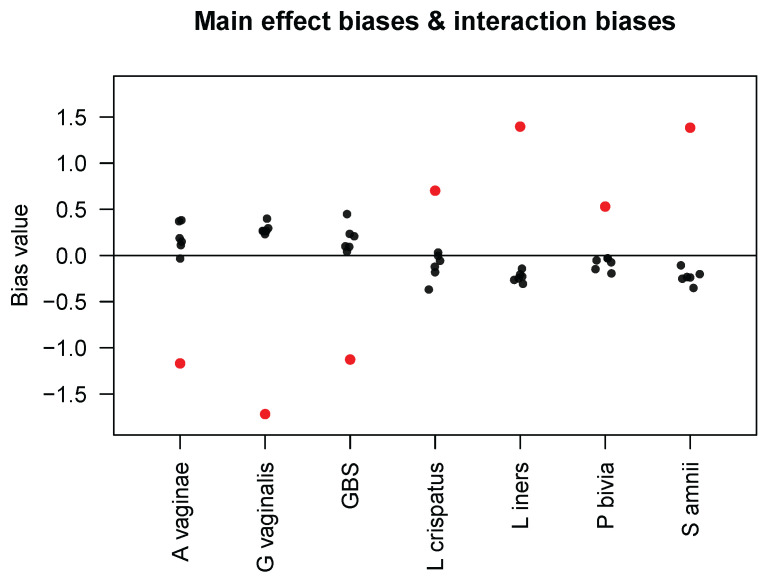
Main effect biases (red dots) of the seven taxa in the Brooks mock community samples and interaction biases (black dots) that each taxon (in the *x*-axis) caused to the other six taxa, as estimated by Zhao and Satten [6].

**Figure 2 genes-14-01777-f002:**
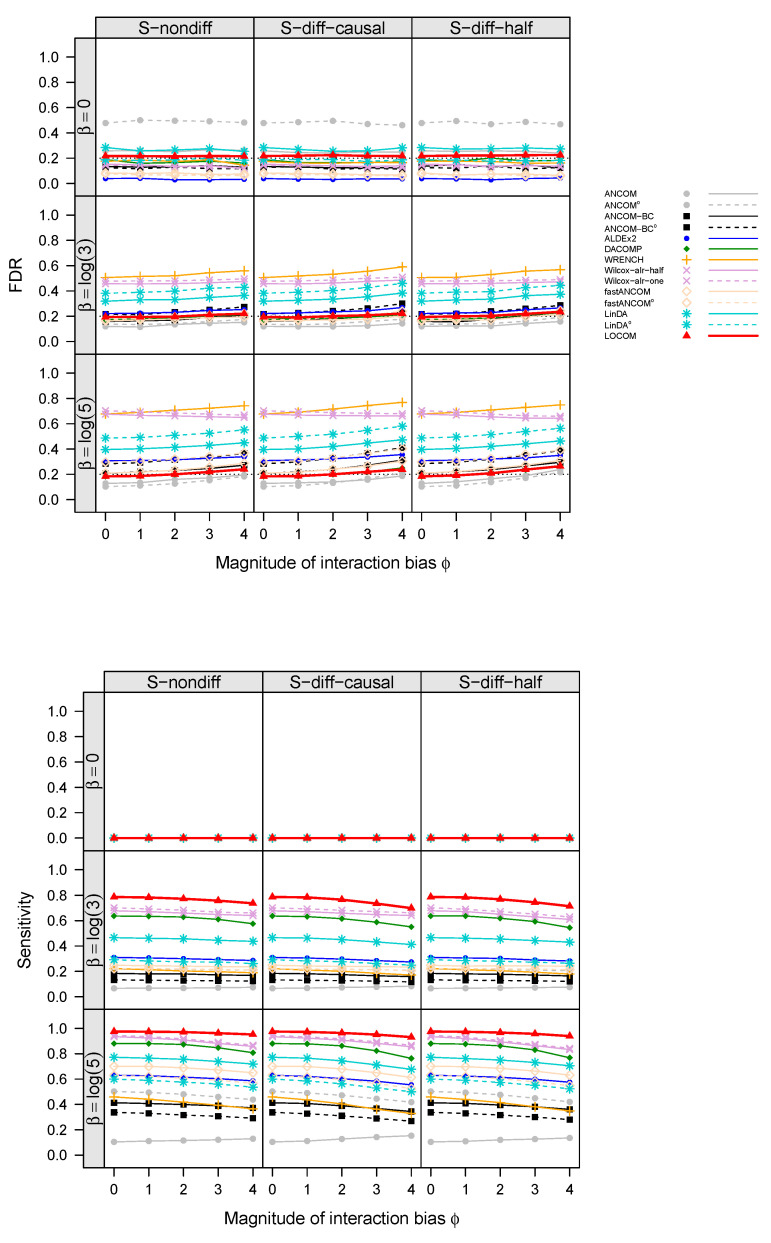
FDR and sensitivity at the nominal FDR 20% (black dotted line) of LOCOM and other compositional methods for data simulated under M1 and with a binary trait. With superscript *^o^*, the filter for rare taxa that are present in less than 10% of samples as adopted by the original programs was used; without *^o^*, the more stringent filter with 20% cutoff as recommended by LOCOM was used.

**Figure 3 genes-14-01777-f003:**
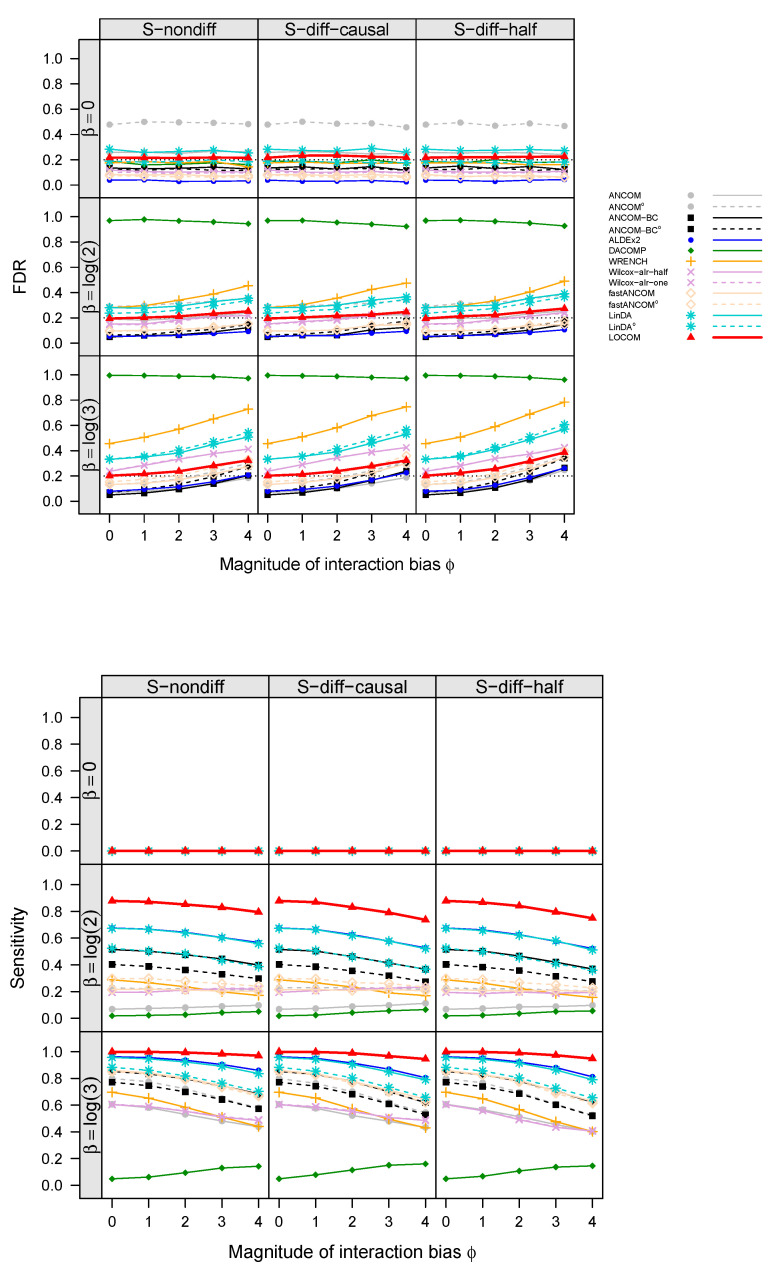
Similar to Figure 2 except that the data were simulated under M2.

## Data Availability

All simulation code are available on GitHub at https://github.com/yijuanhu/LOCOM (accessed on 7 September 2023).

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
