# Peer review of "Impact of Experimental Bias on Compositional Analysis of Microbiome Data"

_genes, 2023, doi:10.3390/genes14091777_

Round 1
Reviewer 1 Report
See attached

Author Response
- Re Reviewer 1’s Comment 1: LOCOM has been designed for testing differential ‘absolute’ abundances, not ‘relative’ abundances. The effect size parameters bj characterize differences in ‘absolute’ abundances rather than ‘relative’ abundances, because there exists a normalization factor ai in the equation (2) of the LOCOM manuscript that is a function of bj For this reason, LOCOM is in the same camp as ANCOM, ANCOM-BC, fastANCOM, and LinDA, all of which, as the reviewer pointed out, are inherently tailored for differential absolute abundance assessment. We have added this clarification to page 4 below Equation (1).
- Re Reviewer 1’s Comment 2: Since CORNCOB and LEfSe have been specifically crafted for differential relative abundance analysis, we decided not to include them in our evaluation, which was restricted to methods for differential absolute abundance analysis. We have uploaded the simulation code ‘sim_binaryY_bias.R’ to the ‘simulation_interaction_bias’ directory of our Github site.
Reviewer 2 Report
Summary: The manuscript addresses the issue of experimental bias in microbiome data analysis, which is often overlooked in the development of statistical methods. Taking into consideration taxon-taxon interactions, the authors formulated an interaction bias model and conducted simulations to evaluate the performance of the proposed method, LOCOM. The results demonstrate that LOCOM remains robust within a reasonable range of interaction biases.
The manuscript provides valuable insights into addressing experimental bias and interaction effects in microbiome data analysis, contributing to the advancement of statistical methodologies in this field.
Comments:
1、LOCOM is an association analysis method designed to address experimental biases and compositional effects in microbiome research. However, it is important to note that in the manuscript, the term "causal taxa" is used to describe taxa that are associated with the trait variable. This terminology could inadvertently lead readers to infer a causal relationship between these taxa and the trait.
2、A common practice for controlling the false discovery rate (FDR) is to set the threshold at 0.05. In the manuscript, a higher threshold of 0.2 is used. Could this higher threshold potentially raise concerns?
3、In the Supplementary Materials, a comprehensive explanation is provided regarding the generation of simulated data. The read counts for 856 taxa were generated using a multinomial distribution, with a mean sample size of 10,000. This methodology results in highly sparse observational data. It would be valuable if you could elaborate on the specific steps taken during your data preprocessing and clarify how LOCOM addresses observations with zero read counts.
Minor editing of English language required.
Author Response
- Re Reviewer 2’s Comment 1: We have added the explanation that ‘causal’ means "associated with the trait" in the first appearance of every section.
- Re Reviewer 2’s Comment 2: We chose a relatively high nominal FDR level (20%) because the numbers of causal taxa in both M1 and M2 were small. We have added this explanation to the end of the Methods section (page 8). In the previous LOCOM paper, we also considered a lower nominal FDR level of 10% and LOCOM still controlled FDR at this level.
- Re Reviewer 2’s Comment 3: LOCOM requires that taxa present (i.e., having non-zero counts) in fewer than 20% of samples are filtered out. Since LOCOM fits logistic regression models to taxa count data, zero read counts are naturally handled as possible outcome values. We have added this statement to the top of page 8.